# Hydrodynamics and Mass Transfer in an Airlift Loop Reactor: Comparison between Using Two Kinds of Spargers

**Xiao Xu \* and Yingchun Zhang**

School of Mechanical and Power Engineering, East China University of Science and Technology, Shanghai 200237, China; ecust_zyc@163.com
\* Correspondence: xxu@ecust.edu.cn

**Abstract:** The effects of different spargers on the hydrodynamics and mass transfer of an airlift loop reactor were investigated. The gas holdup, liquid loop velocity, and volumetric mass transfer coefficient of the reactor were tested using a ring orifice distributor and a jet nozzle. The study was conducted in a 6 m high airlift loop reactor at a superficial gas velocity of 0.01~0.04 m/s, and the superficial liquid velocity was maintained at 0.0154 m/s. The results showed that using the jet nozzle provided a higher gas holdup, liquid loop velocity, and mass transfer. When the superficial gas velocity was less than 0.0325 m/s, the liquid loop velocity generated by the jet nozzle was approximately 1.1-fold higher than that generated by the ring orifice distributor, and the disparity in gas holdup between the riser and downcomer enhanced the power of liquid circulation. When the superficial gas velocity was more than 0.0325 m/s, the jet kinetic power dominated the improvement in the liquid loop velocity, and the energy input from the nozzle to the airlift loop reactor was greater than 10.8 J/(s·m$^2$). This indicated a threshold of energy input for overcoming the friction loss. In this situation, the liquid loop velocity in the jet form increased considerably, thus favoring the mixing performance and temperature uniformity of the reactor. It was also of significance for avoiding the formation of a flow dead zone in scale-up airlift loop reactors.

**Keywords:** gas holdup; liquid loop velocity; mass transfer; airlift loop reactor; ring orifice distributor; jet nozzle





## 1. Introduction

Bubble columns are two-phase or multiphase reactors where a gas phase is dispersed in the form of "non-coalescence-induced" or of "coalescence-induced" bubbles. The two phases are separated by an interface between the dispersed phase and the continuous phase; at this interface, the transfer of heat and mass may occur [1]. Bubble columns are widely used in chemical, petrochemical, and biochemical industries because of their high gas–liquid mass transfer rates, easy operation and maintenance, and their ability to deal with a wide range of flow rates and reaction conditions. The column design and operating conditions can be modified to optimize performance for specific applications. There are several types of bubble columns, including conventional, airlift, and draft tube bubble columns.

An airlift reactor is a type of multiphase reactor that uses gas–liquid circulation induced by gas sparging to promote mixing, mass transfer, and heat transfer [2]. The unique feature of airlift reactors is that they employ two distinct regions of liquid flow: an upflowing region and a downflowing region [3]. The upflowing region is formed by the gas–liquid mixture, whereas the downflowing region is formed by the liquid that is displaced by the rising gas bubbles. Airlift reactors are widely used in many chemical and biochemical processes, including wastewater treatment, fermentation, and bioreactor design [4–6]. Multiphase hydrodynamics in airlift reactors play a significant role in determining their performance [7]. The interaction between gas bubbles and the liquid phase influences the

bubble size distribution, bubble rise velocity, and gas holdup in the column. The bubble size distribution affects mass transfer and mixing, while the bubble rise velocity controls the gas holdup and circulation rate [8,9]. Therefore, the accurate prediction of the hydrodynamics in airlift reactors is critical for optimizing performance and designing better reactors.

To date, numerous studies regarding the fluid characteristics and mechanism of gas–liquid mass transfer in the airlift loop reactor have been reported. Many researchers have focused on investigating the performance of airlift loop reactors by studying various factors. These factors include the operating conditions, viscosity, and surface tension of the experimental system, as well as the size and structure of the reactor [10,11]. Specifically, the influence of the gas distributor on the column performance has been a subject of research, with significant attention given to the structure and design of the distributor [12,13]. The two most commonly studied types of gas distributors are the bubble disk and the ring. The design of the gas distributor plays a crucial role in determining the boundaries of the flow regime, indicating that the initial bubble size has a significant impact on the gas holdup ($\varepsilon$) and mass transfer [14]. The bubble-type gas distributor limits the range of diameters for the generated bubbles. The initial sizes are mostly 5~25 mm [15], and the bubble diameters are large, which limits research regarding the influence of bubble size on gas–liquid mass transfer. The gas–liquid mixing jet nozzle, generating smaller bubbles, provides the gas–liquid feeding mode of the column. The effects of the smaller bubbles and jet energy on the hydrodynamics should be studied.

The differential pressure method is used to measure the pressure at different heights of the reactor bed. $\varepsilon$ is obtained by calculating the decline in pressure between the measurement positions. This method is widely used because of its advantages, such as convenient measurement, high accuracy, and good sensitivity [16,17]. The probe method is efficient and highly accurate, is widely used in studying bubble motion, and is primarily utilized for local $\varepsilon$ measurements [18]. The liquid loop velocity, $U_L$, is another critical parameter of the airlift loop reactor. The tracer method is commonly employed to calculate the liquid velocity by measuring the times at which the tracer passes through two probes at certain positions. The common tracers are saturated salt [19] and acid–base solutions [20], etc. $k_L a$ indicates the quality of the mass transfer effect of the gas–liquid reactor. In $k_L a$, the liquid-side mass transfer coefficient $k_L$ is a function of liquid turbulence [21], and $a$ is the specific surface area of the gas–liquid interface. The main measurement method is the dynamic oxygen concentration method [22]. Cerri et al. [23] and Wongsuchoto et al. [24] suggest that the contribution rate of $a$ in increasing $k_L a$ is higher than that of $k_L$. The method to increase $a$ involves decreasing the bubble diameter and increasing the $\varepsilon$ of the reactor, whereas increasing $k_L$ requires the enhancement of liquid turbulence.

The objective of this study is to analyze fluid parameters $\varepsilon$, $U_L$, and $k_L a$ in a 6 m airlift loop reactor and compare the performances of two sparger configurations, namely the ring orifice distributor and the jet nozzle. The study aims to identify the reasons behind the observed increase in $U_L$ and evaluate the overall performance of the reactor.

## 2. Experimental and Methods

### 2.1. Experimental Setup

The airlift loop reactor consisted of outer and inner cylinders. The outer cylinder displayed an inner diameter and a wall thickness and height of 280, 10, and 6100 mm, respectively. The inner cylinder exhibited an inner diameter and a wall thickness and height of 200, 5, and 5600 mm, respectively. The height/diameter ratio H/D was 21.8. The annulus channel and the inner cylinder served as the downcomer and the riser. A riser and downcomer with cross areas of 0.028 m$^2$ and 0.030 m$^2$, respectively, were formed as the bubbles were injected into the inner cylinder. The riser was a vertical tube where gas and liquid were mixed to form gas–liquid dispersion bubbles. When the gas was injected into the riser, it generated a positive pressure that drove the liquid flow. The upward flow carried the entrained gas–liquid mixture to the top of the reactor. The liquid–gas mixture, after being separated in the upper portion of the vessel, flowed down through the

downcomer by the effect of gravity. Both cylinders consisted of transparent plexiglass to enable the observation of the experiments.

Six threaded holes were generated in the wall of the outer cylinder for the connection of external pressure sensors to measure gas holdup in the downcomer. The gas holdup in the riser was measured using two immersed pressure sensors. With the sensor installation positions as the dividing points, the column was divided into zones A, B, C, D, and E from the top to the bottom along the height. Two conductivity probes to detect the tracer and a tracer injector to provide the tracer were also set into the wall of the outer cylinder. The locations of the sensors and the process flow diagram are shown in Figure 1.

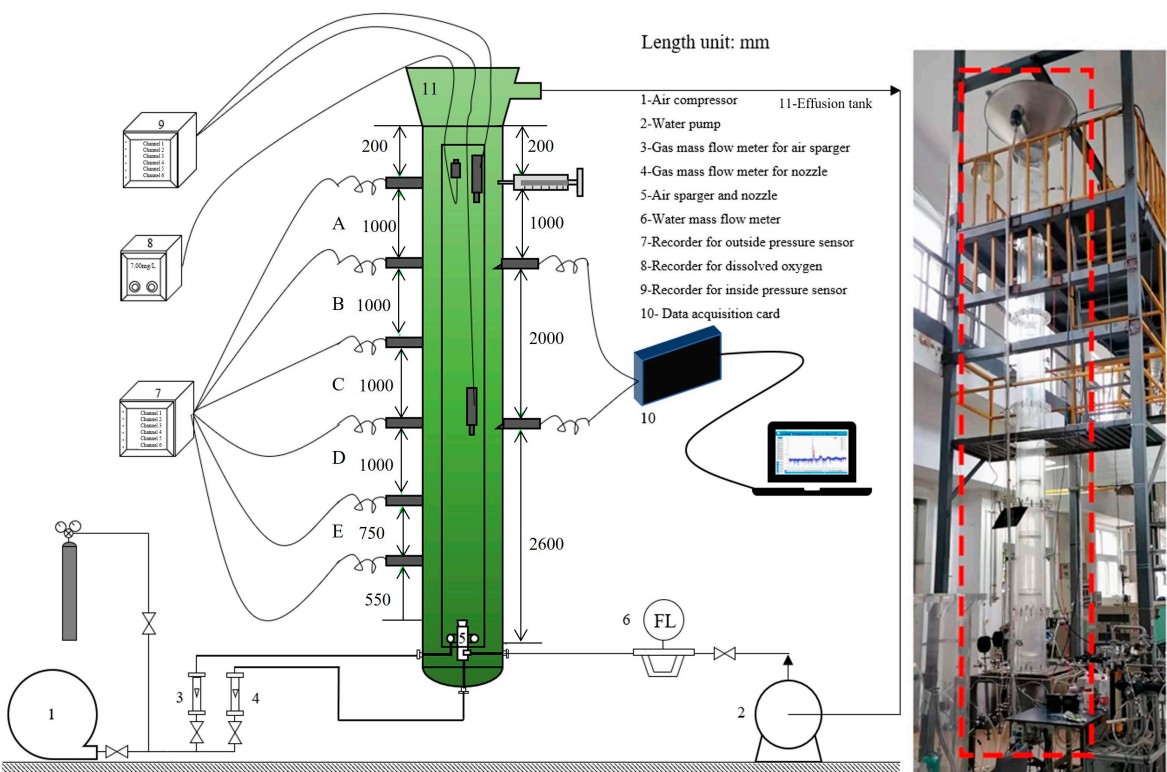

**Figure 1.** Process flow diagram and the airlift loop reactor.

Initially, deionized water was introduced into the column, gradually filling it until the liquid level reached the effusion tank. The water pump, with its inlet connected to the liquid outlet of the effusion tank, was then switched on, and the pumped water passed through the mass flow meter and liquid inlet of the column. The pumped water flow rate was 1.74 $m^3$/h. The superficial liquid velocity was maintained at 0.0154 m/s. Subsequently, the pressurized water entered the jet nozzle, with the liquid mass flow into the nozzle maintained at 1.74 $m^3$/h. The gas entered the ring orifice distributor or jet nozzle, depending on the mode used. The $U_g$ in the riser was set as 0.01, 0.0175, 0.025, 0.0325, or 0.04 m/s, and finally, the bubble groups were well mixed above the bubble generators. As a result, the swarm of bubbles and liquid moved upward, and the gas spilled into the atmosphere through the gas outlet in the effusion tank. Due to the decrease in the mixing density caused by gas entrainment in the riser, the liquid within the downcomer displayed a continuous downward trend. The liquid flowed upward and downward to generate liquid circulation within the column. The airlift loop reactor was operated in batch mode under all experimental conditions.

### 2.2. Bubble Generation

In this study, two gas spargers were inserted at the bottom of the bubble column separately and used to control the initial bubble size. One was an annular tube with

39 evenly distributed holes with diameters of 2 mm, which denoted the ring orifice distributor, and the detailed structure of the annular tube is shown in Supplementary Materials (Figure S1). The other gas sparger, which denoted the jet nozzle, was a nozzle with a tangential fluid inlet and circular orifice jet. As reported in previous studies, the jet nozzle produced smaller bubbles [25–27]. The detailed structure of the jet nozzle is shown in Supplementary Materials (Figure S2), and the size of the nozzle is shown in Table 1.

**Table 1.** Size parameters of the jet nozzle generator.

| $d_{m,c}$/mm | $h_{m,c}$/mm | $d_{G,in}$/mm | $d_{L,in}$/mm | $d_{G,out}$/mm | $d_{L,out}$/mm |
|---|---|---|---|---|---|
| 26 | 50 | 2.5 | 8 | 8 | 8 |

In the ring orifice distributor, the gas was injected into the annular tube, whereas none of the liquid was injected, which was defined as the bubbling form. In the jet nozzle, the gas and liquid were mixed and injected into the nozzle, which was defined as the mixed-jet form. The liquid injected into the nozzle was from the bottom of the column.

The ring orifice distributor and jet nozzle generated bubbles with varying initial diameters. Numerous studies in literature have explored the size calculations of bubbles produced by both the nozzle and the bubble distributor in an airlift loop reactor. The correlation formula of Cramers et al. [28]. is especially more accurate in predicting the diameters of bubbles generated by jet nozzles.

$$d_{N,nozzle} = 2.64 \frac{\sigma_L^{0.6}}{\rho_L^{0.4} \rho_G^{0.2} \left(\frac{P}{V}\right)^{0.24}} \frac{1 + \varphi_G}{1 + 0.2\varphi_G} \tag{1}$$

$$\varphi_G = \frac{V_{G,nozzle}}{V_{G,nozzle} + V_{L,nozzle}} \tag{2}$$

Power per volume can be estimated from the liquid kinetic power in the nozzle and the mixing shock volume, which can be approximated as being the same as the nozzle volume [29]. Hence, we obtained:

$$\frac{P}{V} = \frac{1}{2000L} \rho_L U_{L,nozzle}^3 \tag{3}$$

Bhavaraju et al. [30] provided a correlation equation for the initial bubble diameter of the ring orifice distributor, and it showed that initial bubble size varied with surface tension, viscosity, orifice diameter, and superficial gas velocity. It was shown that:

$$\frac{d_b}{d_0} = 3.23 \mathrm{Re}_L^{-0.1} Fr_L^{0.21} \tag{4}$$

*2.3. Measurements*

2.3.1. Gas Holdup Measurement

The gas holdups of the riser and downcomer were measured. The riser gas holdup was measured using immersed pressure sensors, as shown in Supplementary Materials (Figure S3), and the downcomer gas holdup was measured using external pressure sensors. Two external pressure sensors located at the top and bottom of the downcomer, separated by 4750 mm, measured the total gas holdup of the downcomer. Two immersed pressure sensors located at the top and bottom of the riser, separated by 3000 mm, measured the total gas holdup of the riser. In addition, the local gas holdups in five height zones (A, B, C, D, and E) between adjacent sensors were measured. The pressure sensors were used to measure the total volume fraction of gas, with a measurement error of <2%. The pressure sensors, with 4–20 mA signals, were connected to the data recorder at a recording frequency of 1 Hz. The recording time of the pressure under each mode was >200 s, and the average and deviation values were recorded to calculate gas holdup and the relative error. The gas

holdup could be calculated based on the difference in the mean pressure with or without gas flow at a certain height. In a previous study, similar gas holdup measurements were verified [26]. Measuring gas holdup using a differential pressure transducer is a standard procedure, as described in previous studies [31,32].

The gas holdup values of the riser and downcomer were calculated according to Equation (5). The airlift loop reactor total gas holdup $\varepsilon_g$ was calculated using Equation (6) [33], according to the riser total gas holdup $\varepsilon_r$, downcomer total gas holdup $\varepsilon_d$, cross areas of the riser $A_r$, and downcomer $A_d$.

$$\varepsilon = \frac{\Delta P_1 - \Delta P_2}{\Delta P_1} \tag{5}$$

$$\varepsilon_g = \frac{\varepsilon_r A_r + \varepsilon_d A_d}{A_r + A_d} \tag{6}$$

Here $\Delta P_1$ and $\Delta P_2$ are the differences in the mean pressure without and with gas flow at a certain height, respectively.

### 2.3.2. Liquid Loop Velocity Measurement

The tracer method was used to measure the liquid loop velocity in this study, with two conductivity probes separated by 2 m fixed into the wall of the outer cylinder to detect the tracer. The structure of one of the conductivity probes is shown in Supplementary Materials (Figure S4a), and a 20 mL saturated NaCl solution was used as the tracer. As the tracer flowed around the conductivity probe, the electric current signal from the recorder for conductivity probes surged. When the tracer circulated a complete cycle, the time interval from the peak signal of one conductivity probe to that of the same conductivity probe represented the time required for the liquid to flow through the riser twice, and thus, the liquid loop velocity could be calculated. The time from the peak signal of one probe to that of the other probe represented the time required for the liquid to flow a distance of 2000 mm. The mean liquid velocity within the downcomer could then be calculated.

The calculation of liquid loop velocity requires the time difference between the two peaks. In this study, the moving average filtering method was used to filter noise and yield two relatively smooth curves, which could be used to accurately identify the corresponding time of each peak. The originally acquired and filtered signal curves are shown in Supplementary Materials (Figure S5), using a $U_g$ of 0.01 m/s in the riser as an example. After filtering, the peak of the signal was clear. Matlab (MathWorks, Natick, MA, USA) was used to identify the coordinate of the peak point, where the x-coordinate was the equivalent conversion form of the sampling time. In Matlab, the dateaxis function was used to convert it into the form of h:m:s, and then h:m:s was mapped to the sampling point in the original data. This method accurately recorded the sampling time to the millisecond level, based on the accuracy of the data acquisition card.

The liquid loop velocity $U_L$ was calculated using the adjacent peaks of a single probe. Assuming that the corresponding time difference between the two peaks of the same probe was $\Delta T$, and the inner cylinder height was 5.6 m, the average $U_L$ between the riser and downcomer was calculated as follows:

$$U_L = \frac{2 \times 5.6}{\Delta T} \tag{7}$$

The liquid velocity in the downcomer $U_{Ld}$ was calculated using the adjacent peaks of both probes, separated by a vertical distance of 2 m. According to the peak time difference$\Delta t$, the equation used for calculating $U_{Ld}$ was:

$$U_{Ld} = \frac{2}{\Delta t} \tag{8}$$

2.3.3. Volumetric Mass Transfer Measurement

The dynamic oxygen concentration method was used to measure the volumetric mass transfer coefficient values. This method requires a dissolved oxygen sensor with a much shorter response time, compared to that of the mass transfer of the system. The response time of the dissolved oxygen sensor used in this study was <15 s, which enabled the accurate measurement of the volumetric mass transfer coefficient. The liquid circulation process remained unchanged. First, $N_2$ gas was introduced until the dissolved oxygen concentration was reduced to 0.3 mg/L, which was <7% of the saturated dissolved oxygen concentration. The $N_2$ cylinder was then closed, and the air compressor intake was opened. The increasing dissolved oxygen concentration was recorded until it reached the saturation concentration. The relationship between the dissolved oxygen concentration and time was then plotted. The liquid was well mixed, and the CSTR model was adopted. According to the continuous stirred-tank reactor (CSTR) model, there was a relationship between the concentration and the mass transfer coefficient, as shown in Equation (9). The volumetric mass transfer coefficient $k_L a$ was obtained by fitting the equation of the mass transfer coefficient to experimental data, which is shown in Equation (10). The error of the mass transfer coefficient measured using this method was <6%. According to the continuous stirred-tank reactor model, the equation used for calculating the mass transfer coefficient at 20 °C was as follows (11):

$$\frac{c^* - c}{c^* - c_{t=0}} = \exp\{-(k_L a)_L t\} \tag{9}$$

$$\ln \frac{c^* - c_0}{c^* - c_L} = k_L a t \tag{10}$$

where $C^*$ is the saturated oxygen concentration of the liquid, and $C_L$ is the oxygen concentration at time $t$. When $t = 0$, $C_L = C_0$.

Considering that the saturated oxygen concentration was related to temperature, a temperature correction was achieved for the volumetric mass transfer coefficient, following [34]. The volumetric mass transfer coefficient converted to 20 °C was finally adopted.

$$(k_L a)_{20°C} = \frac{k_L a_T}{(1.022^{T-20})} \tag{11}$$

where $k_L a_T$ is the volumetric mass transfer coefficient at temperature $T$.

**3. Results and Discussion**

*3.1. Gas Holdup ε*

The total $\varepsilon$ values in the riser and downcomer are shown in Figure 2. With the increase in $U_g$ from 0.01 to 0.04 m/s, the total $\varepsilon_{total}$ generally increased. $\varepsilon_{total}$ in the mixed-jet form was always higher than that in the bubbling form. When $U_g = 0.04$ m/s, the total $\varepsilon_{total}$ in the riser in the mixed-jet form reached approximately 15%, whereas that in the bubbling form was approximately 9%. This was because the initial bubble size produced by the jet nozzle was smaller and more uniform, and smaller bubbles rose slowly in the riser. In addition, when the bubble size in the riser was small, the separation of the bubbles at the gas–liquid separator at the top of the column was challenging, leading to the increased $\varepsilon$. At $U_g > 0.0325$ m/s, the growth rate of $\varepsilon$ in the riser of the airlift reactor with a mixed-jet form slowed down. In the downcomer, $\varepsilon$ started to exhibit a decelerated growth rate from $U_g = 0.025$ m/s. This was because the increased gas velocity intensified the turbulence of the liquid and the probability of bubble coalescence. This rendered overflow into the atmosphere at the gas–liquid separator at the top easier, reducing the number of bubbles entering the downcomer. Figure 3 shows the difference in the total $\varepsilon_{total}$ values in the riser and downcomer. $\varepsilon_r$ in the riser was always higher than that in the downcomer in both intake forms, and the difference increased with increasing $U_g$. The increase in gas holdup disparity favored liquid circulation. It should be noted that there was a circulation loop of water flow from the bottom of the column to the top of the column. The circulation loop of

water may contribute to the increase in the gas holdup, due to the bubble entrainment in the loop.

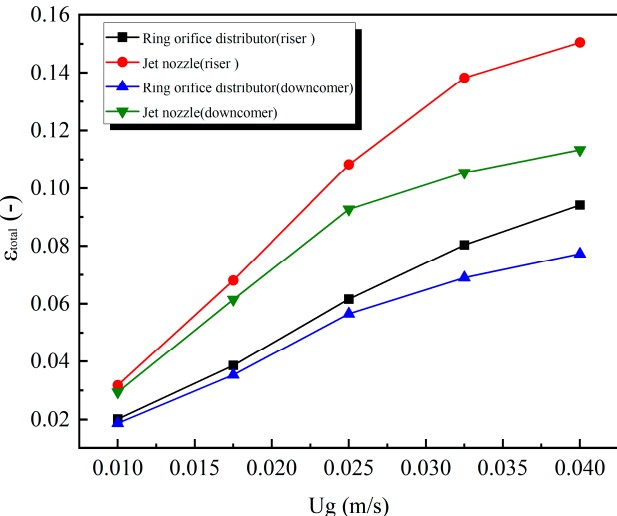

**Figure 2.** Relationships between the total gas holdup $\varepsilon_{total}$ and superficial gas velocities $U_g$ in the riser and downcomer.

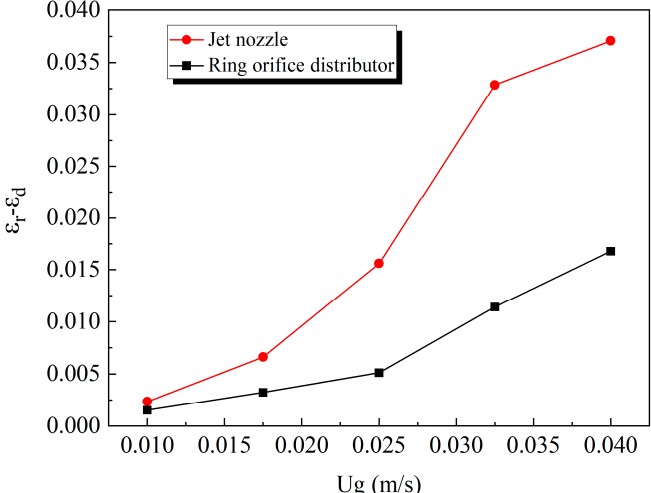

**Figure 3.** Relationships between the disparities in the total gas holdup of the riser $\varepsilon_r$ and downcomer $\varepsilon_d$ and superficial gas velocity $U_g$.

Figures 4 and 5 show the relationships between the local gas holdup $\varepsilon_{rlocal}$ in the riser and the downcomer in zones A and E with $U_g$, respectively. In different zones, the $\varepsilon_{rlocal}$ increased with increasing $U_g$. At $U_g > 0.0325$ m/s, the growth rate of the $\varepsilon_{rlocal}$ in riser zone E decreased in the bubbling form, whereas this phenomenon occurred in both riser zones A.

According to Formulas (1)–(4), the bubble size generated by the ring orifice distributor was about 1.4 mm, while the bubble diameter from the jet nozzle ranged from 0.5 to 0.63 mm. The initial bubble size from the ring orifice distributor was larger than that from the jet nozzle. Consequently, in the bubbling form, due to the larger initial bubble size and lower bubble density, coalescence occurred in the vicinity of the ring orifice distributor zone. However, the bubbles generated by the jet nozzle were small and uniform. As they ascended in the riser, they coalesced at the top, forming larger bubbles to expedite overflow from the gas–liquid separator. Furthermore, the annular Reynolds number of the fluid flow was $Re_a = \frac{D \cdot U_L}{\nu}$. The hydraulic diameter D was a measure of the effective diameter of the annular flow area. There was a variation in liquid loop velocities $U_L$ with superficial gas

velocity $U_g$. The annular Reynolds numbers for fluid flow in the downcomer using a jet nozzle and ring orifice distributor were $5.76 \times 10^4 \sim 1.32 \times 10^5$ and $5.45 \times 10^4 \sim 6.3 \times 10^4$.

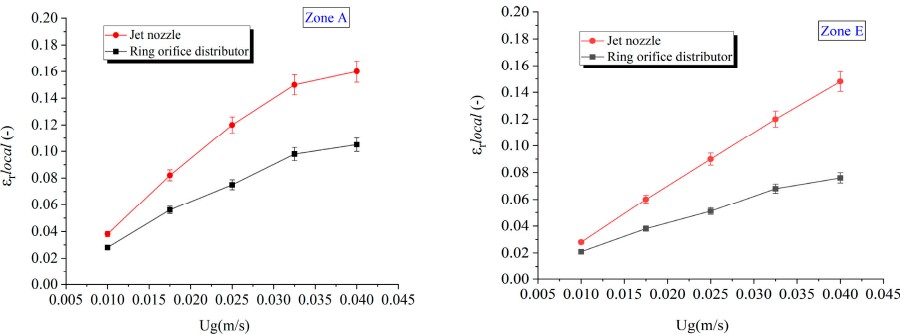

**Figure 4.** Relationships between the local gas holdup $\varepsilon_{\mathrm{r}local}$ and superficial gas velocities $U_g$ in zones A and E of the riser.

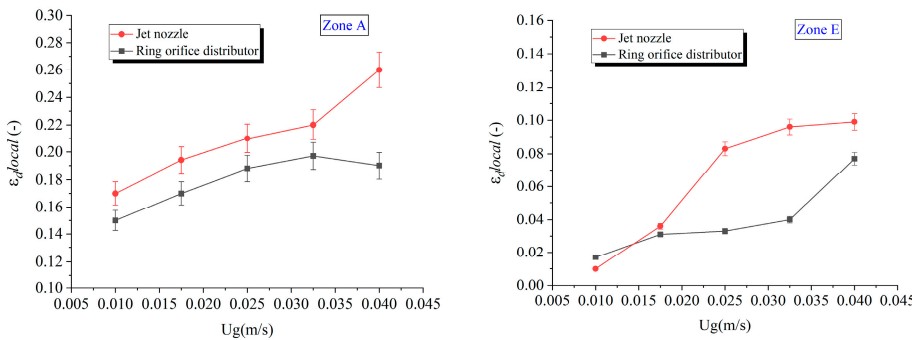

**Figure 5.** Relationships between the local gas holdup $\varepsilon_{\mathrm{d}local}$ and superficial gas velocities $U_g$ in zones A and E of the downcomer.

The distribution of the $\varepsilon_{\mathrm{d}local}$ in the downcomer differed from that in the riser. At $U_g > 0.0325$ m/s, the growth rates of the $\varepsilon_{\mathrm{d}local}$ values in downcomer zone A decreased in the bubbling form. Due to the wall effect of the downcomer, bubbles coalesced and ruptured in the middle and upper sections of the downcomer. When $U_g$ was high, the coalescence and rupturing of bubbles reached equilibrium, and the growth rate of $\varepsilon$ slowed. In the mixed-jet form, as the initial bubble size was small, numerous bubbles entered the downcomer after migrating to the top. The growth rates of the $\varepsilon_{\mathrm{d}local}$ values in downcomer zone E slowed, due to bubble coalescence and overflow. Zones B~D had similar laws to the ones above, and the specific relationship diagram is shown in Supplementary Materials (Figure S6 and S7).

### 3.2. Liquid Loop Velocity $U_\mathrm{L}$

The liquid loop velocity $U_\mathrm{L}$ and downcomer liquid velocity $U_\mathrm{Ld}$ were measured using single and double probes, respectively. Figure 6 illustrates the changes in $U_\mathrm{L}$ for the airlift loop reactor with $U_g$, and the superficial liquid velocity was maintained at 0.0154 m/s. It was found that $U_\mathrm{L}$ increased with increasing $U_g$. In the bubbling form, when $U_g$ was 0.04 m/s, $U_\mathrm{L}$ was only 0.263 m/s. The driving force for liquid circulation was derived from the disparity in the $\varepsilon$ values in the riser and downcomer. Due to the ring orifice distributor generating the large bubbles and their rapid rising velocity, the disparity in the $\varepsilon$ values was small. Thus, the growth rate of $U_\mathrm{L}$ in the bubbling form was not obvious.

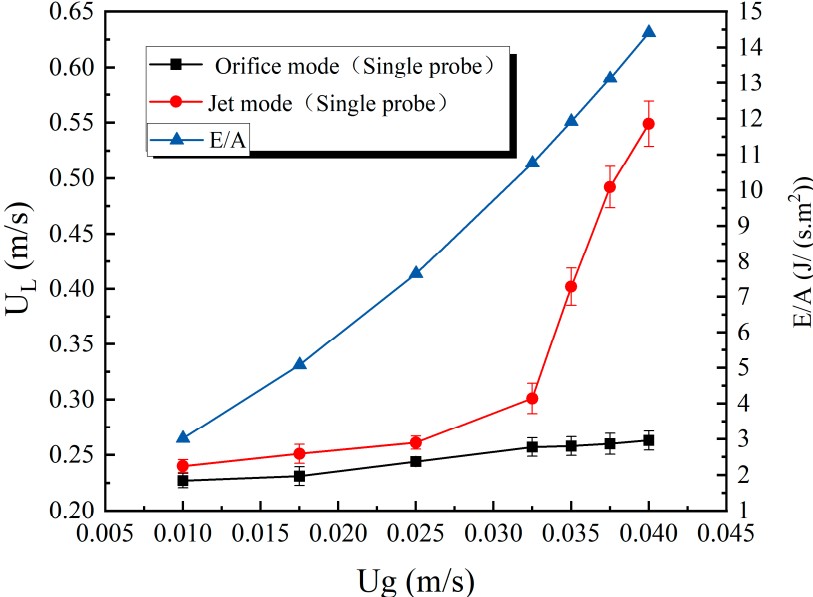

**Figure 6.** Variations in the liquid loop velocities $U_L$ with the superficial gas velocity $U_g$.

In the mixed-jet form, the energy for liquid circulation also came from the kinetic energy of the gas–liquid jet. As the jet kinetic power increased, the friction loss also increased, due to the increase in shear stress between the gas–liquid phases and the reactor walls. To optimize the jet kinetic power, it is essential to balance the momentum of the gas–liquid mixture with the frictional losses encountered in the reactor. The kinetic energy of a jet nozzle can be calculated using the following formula:

$$v_n = (Q_L + Q_G)/A_{nozzle} \tag{12}$$

$$E = \frac{1}{2}(m_L + m_G)v_n^2 \tag{13}$$

Therefore, the jet kinetic powers at the jet nozzle were approximately 32, 53.8, 81.1, 114.1, 126.4, 139.2, and 152.7 J/s at $U_g$ values of 0.01, 0.0175, 0.025, 0.0325, 0.035, 0.0375, and 0.04 m/s, respectively. The growth trend of $U_L$ in the mixed-jet form was divided into two sections. The figure shows that the curve of $U_g$ increasing with $U_L$ can be divided into two sections, with $U_g = 0.0325$ m/s serving as the critical point. At this point, the jet kinetic power was 100 J/s. Liquid circulated through the inner and outer walls of the inner cylinder and the inner wall of the outer cylinder, respectively, with a combined contact area of 10.6 m². Below the critical point, $U_L$ showed a slight increase with $U_g$, but when it exceeded the critical $U_g$, $U_L$ increased significantly. Thus, when E/A exceeded 10.8 J/(s·m²), the energy input from the nozzle to the airlift loop reactor exceeded the friction loss, resulting in a significant increase in the liquid loop velocity within the reactor. According to reference [1], the upper and lower regions of the inner cylinder mainly contribute to friction losses, which can be overcome by the energy sprayed from the nozzle.

Figure 7 shows the results of measuring $U_{Ld}$ using double probes. The growth trend was the same as that of $U_L$, but it was higher than that of the average $U_L$. When $U_g = 0.04$ m/s, the $U_{Ld}$ values in the bubbling and mixed-jet forms reached 0.343 and 0.784 m/s, respectively, whereas the $U_L$ values were 0.263 and 0.549 m/s, respectively. This was mainly because the resistances at the positions where the liquid turned at the bottom and top were high during liquid circulation, particularly at the bottom, which consumed a considerable amount of liquid kinetic energy.

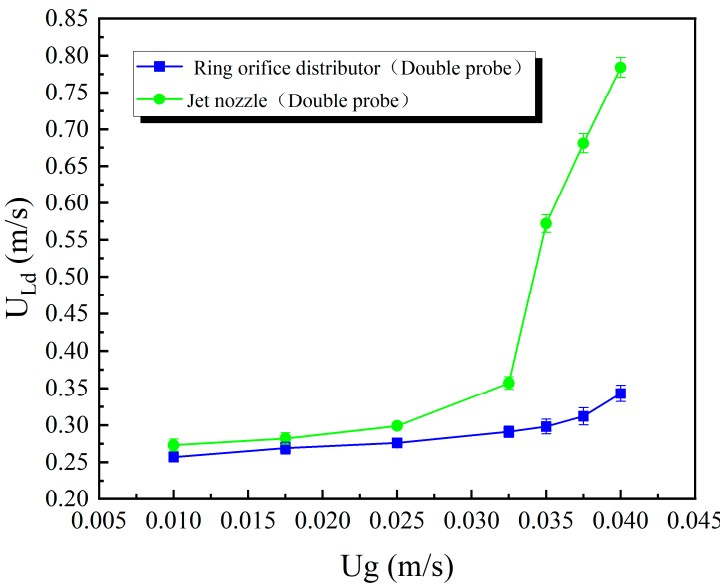

**Figure 7.** Variations in the downcomer liquid velocities $U_{Ld}$ with superficial gas velocity $U_g$.

### 3.3. Volumetric Mass Transfer $k_L a$

Figure 8 shows the change in $k_L a$ of the airlift loop reactor with $U_g$, and $k_L a$ increased with increasing $U_g$. Compared with that in the bubbling form, $k_L a$ in the mixed-jet form exhibited clear advantages. When $U_g$ increased from 0.01 to 0.04 m/s, $k_L a$ in the mixed-jet form increased from 0.00962 to 0.03772 s$^{-1}$, whereas that in the bubbling form increased from 0.0049 to 0.02619 s$^{-1}$. In the mixed-jet form, at $U_g > 0.025$ m/s, the growth rate of $k_L a$ decreased slightly. According to Formulas (1)–(4), the bubble size generated by the ring orifice distributor was about 1.4 mm, and the bubble diameter generated by the jet nozzle was 0.5~0.63 mm. The jet nozzle produced smaller bubbles, and at a certain $U_g$, several small bubbles coalesced and formed large bubbles. The disordered motion of the larger bubbles produced eddy currents, which accelerated the renewal rate of the gas–liquid interface. When $U_g$ was higher, the flow field in the column changed from the transitional to the heterogeneous flow state, the number of large bubbles increased, and the increase in the vortex frequency reduced the gas holdup. This is unfavorable for mass transfer.

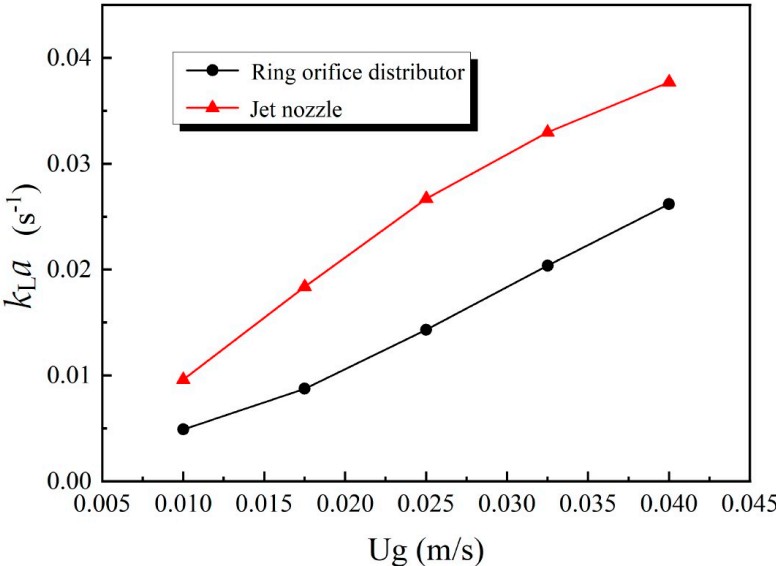

**Figure 8.** Variations in the volumetric mass transfer coefficients $k_L a$ with superficial gas velocity $U_g$.

### 4. Conclusions

This paper shows that the liquid loop velocity of an airlift reactor can be quickly increased when the energy input into the reactor by using the jet nozzle is higher than 10.8 J/(s·m$^2$) at a higher gas holdup. This is a guideline to avoid the formation of a flow dead zone in large-scale airlift loop reactors.

The following conclusions can be drawn from this study:

(1) With increasing $U_g$ from 0.01 to 0.04 m/s, the gas holdup generally increased. The total gas holdup in the riser was notably greater than that in the downcomer for both types of gas spargers, and the disparity in $\varepsilon_{total}$ increased as the $U_g$ increased. The gas holdup disparity was the driving force for the formation of liquid circulation in the reactor, so the increased disparity in $\varepsilon$ favored liquid circulation. The gas holdup of the mixed-jet form was almost always higher than that of the bubbling form. The bubbles generated by the jet nozzle were smaller and uniform, with less overflow, and thus, the jet nozzle displayed more advantages in terms of gas holdup.

(2) The liquid loop velocity increased with the increase in $U_g$. The $U_L$ in the mixed-jet form was always higher than that in the bubbling form. Especially after $U_g$ exceeded 0.0325 m/s, the $U_L$ in the mixed-jet form increased significantly, far more than that in the bubbling form. Because the power of liquid circulation in the reactor in the mixed-jet form came from the disparity in gas holdup on the one hand and kinetic energy of gas–liquid jet on the other, it helped to improve $U_L$. When $U_g > 0.0325$ m/s, the ratio of jet kinetic power to the contact area between the circulating liquid and the reactor was greater than 10.8 J/(s·m$^2$), and the energy input from the nozzle to the airlift loop reactor exceeded the friction loss; the gas holdup disparity was also large, which significantly improved the liquid loop velocity in the reactor.

(3) The volumetric mass transfer $k_La$ increased with increasing $U_g$, and in the mixed-jet form, it exhibited clear advantages. The $k_La$ values exhibited positive linear correlations with $U_g$ in both forms. The positive linear correlations were similar to that of gas holdup. A considerable increase in $U_L$ did not lead to a considerable increase in volumetric mass transfer. Hence, the gas holdup dominated $k_La$ when $U_g$ ranged from 0.01 to 0.04 m/s, and the superficial liquid velocity was maintained at 0.0154 m/s.

**Supplementary Materials:** The following supporting information can be downloaded at: https://www.mdpi.com/article/10.3390/pr12010035/s1, Figure S1: Structure of the ring orifice distributor; Figure S2: Structure of the jet nozzle; Figure S3: Immersed pressure sensors; Figure S4: (a) Schematic diagram of the structure of the conductivity probe; (b) Potential signal acquisition interface of measurement resistance; Figure S5: (a) Originally acquired signal curves of the conductivity probes; (b) Moving-average-filtered signal curves; Figure S6: Relationships between the local gas holdup $\varepsilon$rlocal and superficial gas velocities $U_g$ in areas B~D of the riser; Figure S7: Relationships between the local gas holdup $\varepsilon$dlocal and superficial gas velocities $U_g$ in areas B~D of the downcomer.

**Author Contributions:** Conceptualization, X.X. and Y.Z.; methodology, X.X.; formal analysis, Y.Z.; data curation, Y.Z.; writing—original draft preparation, Y.Z.; writing—review and editing, X.X. and Y.Z.; project administration, X.X. All authors have read and agreed to the published version of the manuscript.

**Funding:** The authors express their sincere gratitude to the National Natural Science Foundation of China (21908057, 52025103) for their financial support. This project was also sponsored by the Chenguang Program supported by the Shanghai Education Development Foundation and the Shanghai Mu-nicipal Education Commission (20CG39).

**Data Availability Statement:** Data are contained within the article.

**Conflicts of Interest:** The authors declare no conflict of interest.

## Nomenclature

| | |
|---|---|
| A | contact area between the circulating liquid and the reactor, $m^2$; |
| $A_d$ | cross-sectional area of the downcomer, $mm^2$; |
| $A_{nozzle}$ | cross-sectional area of nozzle outlet, $m^2$; |
| $A_r$ | cross-sectional area of the riser, $mm^2$; |
| $a$ | specific surface area, $m^2/g$; |
| $C^*$ | saturated oxygen concentration of the liquid, mol/L; |
| $C_L$ | oxygen concentration at any time, mol/L; |
| $d_{G,in}$ | diameter of the gas inlet of the jet nozzle, mm; |
| $d_{G,out}$ | diameter of the gas outlet of the jet nozzle, mm; |
| $d_{L,in}$ | diameter of the liquid inlet of the jet nozzle, mm; |
| $d_{L,out}$ | diameter of the liquid outlet of the jet nozzle, mm; |
| $d_{m,c}$ | diameter of the mixing cavity of the jet nozzle, mm; |
| E | jet kinetic power, J/s; |
| $h_{m,c}$ | height of the mixing cavity of the jet nozzle, mm; |
| $k_L$ | liquid-side mass transfer coefficient, $s^{-1}$; |
| $k_L a$ | volumetric mass transfer coefficient, $s^{-1}$; |
| $m_G$ | gas mass flow rate, kg/s; |
| $m_L$ | liquid mass flow rate, kg/s; |
| $Q_G$ | gas volume flow rate, $m^3/s$; |
| $Q_L$ | liquid volume flow rate, $m^3/s$; |
| $T$ | temperature, K; |
| $\Delta T$ | time difference between the two peaks, s; |
| $\Delta t$ | peak time difference, s; |
| $U_g$ | superficial gas velocity, m/s; |
| $U_L$ | liquid loop velocity, m/s; |
| $U_{Ld}$ | downcomer liquid velocity, m/s; |
| $v_n$ | jet velocity at the nozzle exit, m/s; |
| $\varepsilon$ | gas holdup, -; |
| $\varepsilon_d$ | total gas holdup in the downcomer, -; |
| $\varepsilon_g$ | total gas holdup in the airlift loop reactor, -; |
| $\varepsilon_r$ | total gas holdup in the riser, -; |

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
