# Peer review of "Hydrodynamics and Mass Transfer in an Airlift Loop Reactor: Comparison between Using Two Kinds of Spargers"

_processes, doi:10.3390/pr12010035_

Round 1

Reviewer 1 Report

Comments and Suggestions for Authors

The effects of two kinds of spargers, namely a ring orifice distributor and a jet nozzle, on the hydrodynamics and mass transfer of an airlift loop reactor was investigated. The merit of this article is the report of experimental data. The experimental measurement procedures are carefully explained. However, some comments are listed as followings:

 1.         The performances of the ring orifice distributor and the jet nozzle were not compared with those of other spargers, which are important to judge the spargers’ performance.

2.         On Line 300, the notation UL is a typo, and in Figure 9, KLa is a typo too.

Comments on the Quality of English Language

Minor editing of English language required

Author Response

We are very grateful to Reviewers for reviewing the paper so carefully. We have carefully considered the suggestion of reviewers and make some changes.

Reviewer 2 Report

Comments and Suggestions for Authors

The authors present results obtained from two different injectors used for blowing air into an airlift reactor with an aspect ratio of about 22. The authors measured the gas holdup, the overall mass transfer coefficient and the liquid velocity of the airlift reactor. The gas holdup was measured by pressure transducers. An oxygen problem was used to measure the overall mass transfer coefficient. The liquid velocity was measured by means of measuring the time take for a tracer solution to travel between to point measurements in the downcomer. The injectors were an orifice ring and a jet nozzle. The response of the measured parameters to changes in the gas flow rate were observed for both injectors. They found that for superficial gas velocities above 0.0325 m s-1, that there was an increase in the difference of the gas hold up of the riser from the downcomer for the jet nozzle, which also resulted in a significant rise in the liquid velocity in the downcomer. This may have been due to a transition in the flow regime of the airlift for the jet nozzle. However, this had little influence on the mass transfer coefficient, which did not show the same increase at the transition point velocity, as the trend followed that of the gas hold up in both the riser and the downcomer. Further measurements of the bubble size distribution would help to identify the regime.

Line 91: How does this height to diameter ratio relate to the general design recommendations or existing airlift reactors constructed in industrial settings or studied experimentally?

Lines 322-323: The sentence beginning "The jet nozzle may..." Please could you provide a reference or evidence for the assertion about the smaller bubbles coalescing.

Lines 310-312: What is the ratio of the cross-sectional area of the downcomer to the riser? Is the ratio in the velocity the same or is it slightly different. The flow may accelerate as it changes direction at the top of the downcomer, this may be seen by small bubbles being trapped in the vortices that form at the entry to the downcomer. What is the annular Reynolds number of the fluid flow in the downcomer?

Figure 6-7: It may be useful for the reader to understand how the kinetic energy or power of the injected gas varies with superficial gas velocity. Please could you consider using a second horizontal axis for E/A or even a separate table with the values reported in lines 293-294, with a third column for E/A.

Where in Figures 2-7 can you determine where the change in transitional to heterogeneous flow states? How can you relate this to the type of flow without information on the bubble size distribution in the riser and the downcomer?

Comments on the Quality of English Language

Line 30: "Bubble columns is" should be "Bubble columns are".

Line 31: "because of its high" may read better as "because of their high"

Line 32: "and its ability to" may read better as "and their ability to"

Line 114: "superficial liquid velocity keeps at" may read better as "superficial liquid velocity is maintained at"

Line 213: "(6)" should be "(6)."

Lines 257-260: Please rephrase the sentence beginning with "Conversely". Some more words are required to clarify what you mean about coalescence of the bubbles. If it needs a separate sentence then please do so.

Lines 297-299: Please rephrase, as I can not make sense of what is written. How does the liquid flow through the inner and outer walls of the inner cylinder. Is the inner cylinder is not solid?

Line 300: "UL increases significantly" is UL a mathematical symbol?

Line 332: "velocity can be" may read better as "velocity of an airlift reactor can be"

Line 334: "holdup, which is a guideline" may read better as "holdup. This is a recommendation/guideline to"

Author Response

We are very grateful to Reviewers for reviewing the paper so carefully. We have carefully considered the suggestion of reviewers and make some changes. The specific modifications are shown in the coverletter and manuscript.

Reviewer 3 Report

Comments and Suggestions for Authors

The authors address the importance on gas sparger configurations on hydro dynamics and mass transfer study of airlift loop reactor. The study is significant highlighting the important aspects of industrial gas-liquid contractors applications. I recommend this to be published in Journal of Processes after answering/ incorporating following comments.

1. Several studies have been reported extensively in the past on hydrodynamics and mass transfer in airlift loop reactor for different sparger configurations. Authors are advised to highlight novelty of this work.

2. The methodology section is clear, but it would be beneficial to elaborate on the rationale for choosing the specific range of superficial gas velocities and how they relate to practical applications or industrial relevance.

3. It is obvious that increasing the gas velocity will increase the gas holdup or a sparger promoting the generation of smaller bubbles will give higher gas holdup and volumetric mass transfer coefficients, authors should discuss results logically.

Comments on the Quality of English Language

1. The language used is clear, but some sentences are lengthy and could be divided for improved readability. Additionally, consider defining technical terms or acronyms to ensure accessibility for a broader audience.

Author Response

(The authors gave the same response as above.)
